# Alcohol Recognition and Desire to Drink of Extended Alcohol Brand Logos

**DOI:** 10.3390/ijerph191811756

**Published:** 2022-09-17

**Authors:** Polathep Vichitkunakorn, Sawitri Assanangkornchai, Jirawan Jayuphan, Teerohah Donroman, Tagoon Prappre, Monsicha Sittisombut

**Affiliations:** 1Department of Family and Preventive Medicine, Faculty of Medicine, Prince of Songkla University, 15 Kanchanavanich Road, Hat Yai, Songkhla 90110, Thailand; 2Department of Epidemiology, Faculty of Medicine, Prince of Songkla University, 15 Kanchanavanich Road, Hat Yai, Songkhla 90110, Thailand; 3Division of Digital Innovation and Data Analytics, Faculty of Medicine, Prince of Songkla University, 15 Kanchanavanich Road, Hat Yai, Songkhla 90110, Thailand; 4Faculty of Medicine, Prince of Songkla University, 15 Kanchanavanich Road, Hat Yai, Songkhla 90110, Thailand

**Keywords:** alcohol marketing, reaction time, surrogate marketing

## Abstract

Alcohol companies in Thailand have adopted surrogate marketing that uses similar logos on non-alcoholic products. We aimed to assess variations of the alcohol recognition using reaction time and desire to drink among consumers exposed to original logos and modified logos (i.e., black logos, partial logos, logos on non-alcoholic beverages and other merchandise). Participants aged ≥19 years took part in this cross-sectional study. The primary independent variables were types of logos: original logos, modified logos (i.e., black logos, partial logos, logos on non-alcoholic beverages, and logos on other merchandise). An in-house-developed online survey randomly presented the logos. Alcohol recognition and the desire to drink alcohol were assessed. The study included 1185 participants. More time (estimated coefficient of reaction time <0.5 s) was required to recognize the modified logos than the original logos. Younger participants (19–24 years) reacted significantly faster than the older participants (>25 years) after seeing all types of logos. The desire to drink alcohol (<0.5 point) upon seeing the modified logos was lower than the original logos. No significant difference in the desire was observed between the younger and older participants upon seeing the original and partial logos. The modified logos reminded consumers of the alcohol products of that brand with a tiny difference in reaction time and the desire to drink without practical significance.

## 1. Introduction

Alcohol is a threat to the health and well-being of a whole population [1]. Therefore, a reduction in the number of alcohol drinkers is essential. In 2008, Thailand’s total alcohol consumption per capita was 8.3 L, marked as the highest in ASEAN countries [2]. According to the 2021 Health Behavior of Population Survey of Thailand’s National Statistical Office, 46.4% of participants had drunk alcohol at least once. Thus, the estimated number of Thai drinkers was 26.4 million people [3]. A previous study in Thailand estimated 11,887 alcohol-attributable deaths per year [4]. Thailand has a rather strict alcohol marketing control policy. The Alcoholic Beverage Control Act B.E.2551 (the Act), which was issued in 2008, does not allow the advertisement of alcoholic beverages or display of the logo or name of the beverage in a way that can encourage alcohol purchase. Moreover, the restriction on advertisement airing time on television was also put in place [5].

It has been demonstrated that alcohol advertising can lead to purchasing alcohol and the initiation of alcohol use [6,7,8,9]. Therefore, a key factor is tight alcohol advertising regulations. Nonetheless, Thailand’s alcohol advertising in recent years has become more complex and problematic as various tactics were used to utilize the loophole of the Act [10].

One of the strategies which many companies have adopted is brand extension. A company that employs this technique uses a similar alcohol product logo on its other products, which makes it easy for new products to be recognized by its pre-existing customers [11]. Based on this strategy, surrogate marketing has emerged. The surrogate marketing is a type of brand extension where a brand intentionally uses the extended products to promote an established brand which has marketing restrictions [12]. It is often used when the advertisements of some products (e.g., alcohol, tobacco) are banned. The companies then shift to promoting similar-looking brand marks on their non-alcoholic products in the advertisements [13,14]. Brand extension of such products can be seen around the world, such as in Thailand, India [15,16], and Australia [17]. In India, where surrogate advertising is a concern, the government has issued a surrogate advertising restriction policy [18]. In Thailand, well-known examples are drinking water and soda. There are also promotional merchandise and other giveaways with logos printed on them (e.g., soccer T-shirts). A previous study found that seeing alcohol brand logos on different products can lead consumers to recall the company’s main alcoholic product [19]. Another study conducted in Thailand also found that over 67% of the participants could remember the brand logos and 65% recognized the extended products under the same branding [20]. According to Agarwal’s study in India, 81% of research participants agreed that they bought any related products under the influence of surrogate advertisements [21]. The implication is that those non-alcoholic products can serve as advertisements for alcoholic drinks. Moreover, surrogate advertising is often seen on social media platforms (i.e., Facebook, YouTube), which are less strictly regulated [22]. The social media marketing can be driven by influencers and peer communication that can lead to brand promotion and purchase intention [23,24]. This is particularly concerning as studies have found that alcohol-related content on social media was related to alcohol consumption [25,26].

Previous studies [19,27,28,29] were conducted qualitatively and used paper-and-pencil interviews that did not consider the degree of recognition in terms of reaction time, which was defined as the period of time a person took to respond to a stimulus. Therefore, the degree to which these extended brand marks could impact a person is not clearly understood.

From this perspective, a study on the relationship between brand mark extension and the recognition of alcoholic brands using computer-associated self-interviewing (CASI) that can acquire the reaction time will express the degree of the effect. Our study aimed to assess variations in alcohol recognition time (time to recall alcohol-related product) and desire to drink among participants exposed to original logos and modified logos (i.e., black logos, partial logos, logos on non-alcoholic beverages and other merchandise). We hypothesized that there were no differences in brand recognition times and desire to drink when our participants exposed the original logos and modified logos. For policy implication, the policy-makers (i.e., the Office of Alcohol Control Committee, Department of Disease Control, Ministry of Public Health, Thailand) can take this evidence into account when making decisions on total ban of brand mark extension-related policies.

## 2. Materials and Methods

### 2.1. Study Design and Study Setting

This was a cross-sectional study using an online survey conducted in Thailand by a research group from Prince of Songkla University, Songkhla Province. We developed an in-house web-based online survey application specifically for this study based on the CASI system. Respondents who accessed the survey on their personal devices (e.g., computers, smartphones) were presented with questions. They could respond by either selecting the option of their choice or typing their answers.

### 2.2. Study Population and Sample

The population of this study was the general population aged 19 or older. At preparation phase, we used the “estimating an infinite population mean” formula to calculate the sample size. The standard deviation of response time of 2.5 s from Sinclair et al. [30] and a margin of error of 0.2 s were used to determine the target sample size, which was 601 participants for this study. An estimate of 30% of unusable data from a sample of 601 participants was considered. Therefore, the final sample size was 859 participants, which was enough for estimating brand recognition time in our study. Finally, we recruited 1185 participants at the end of our data collection.

### 2.3. Data Collection

We collected data through an online survey by distributing the recruitment message with a link to access our web application on various Facebook pages and groups for approximately six weeks from February to April 2020. We offered a prize draw to increase the number of participants. The self-reported questionnaire included four parts: (1) alcohol recognition using reaction time; (2) desire to drink alcohol (please see the details in Section 2.4); (3) the alcohol, smoking, and substance involvement screening test (ASSIST); and (4) demographic data and other information (please see Questionnaire S1 in the Appendix A for the questionnaire).

To evaluate the levels of risky alcohol use in our volunteers, we adopted the ASSIST questionnaire, which is typically used in clinical settings [31]. It consisted of the following seven questions related to the use of alcohol.

“In your life, have you ever used alcohol?” (Answer: Yes = 0, No = 3).“In the past three months, how often have you used alcohol?” (Answer: Never = 0, Once or twice = 2, Monthly = 3, Weekly = 4, Daily or almost daily = 6).“During the past three months, how often have you had a strong desire or urge to use alcohol?” (Answer: Never = 0, Once or twice = 3, Monthly = 4, Weekly = 5, Daily or almost daily = 6).“During the past three months, how often has your use of alcohol led to health, social, legal, or financial problems?” (Answer: Never = 0, Once or twice = 4, Monthly = 5, Weekly = 6, Daily or almost daily = 7).“During the past three months, how often have you failed to do what was normally expected of you because of your use of alcohol?” (Answer: Never = 0, Once or twice = 5, Monthly = 6, Weekly = 7, Daily or almost daily = 8).“Has a friend or relative or anyone else overexpressed concern about your use of alcohol?” (Answer: No, never = 0, Yes, in the past three months = 6, Yes, but not in the past three months = 3).“Have you ever tried and failed to control, cut down, or stop using alcohol?” (Answer: No, never = 0, Yes, in the past 3 months = 6, Yes, but not in the past three months = 3).

After adding up the overall score, the participants were categorized into three groups: low risk (0–10), intermediate risk (11–26), and high risk (≥27). We gave brief advice and warnings according to the results at the end of the survey.

### 2.4. Dependent Variables—Alcohol Recognition Using Reaction Time and Desire to Drink Alcohol (Part 1 and 2 of Questionnaire)

We examined the alcohol recognition and desire to drink of participants using two parameters: reaction time (part 1) and desire to drink alcohol (part 2). Reaction time in our study was the period of time a participant took to react to this question: “After seeing this image, do you think of alcoholic beverage?” (answer: yes/no). This question was shown at the beginning of the section. The participant was then randomly shown a total of 30 images, one at a time, with a yes or no choice to respond within the maximum of ten seconds per image. The length of time (in seconds) needed to answer (yes/no) was recorded as reaction time. The yes/no answer was recorded as dichotomized data. Each category of logos consisted of products related to the alcohol company (20 images) and unrelated products as the control (10 images). To minimize information bias, the images were randomly selected from the image pool.

Since the ultimate objective of marketing is to have consumers purchase the products, it is necessary to trigger the desire to drink alcohol. To measure the level of a participant’s desire to drink alcohol (part 2), we asked this question: “*After seeing this image, how much do you want to drink alcohol?*” (answer: 0–4, when 0 = do not want at all, 1 = do not want, 2 = neutral, 3 = want to drink, and 4 = want to drink very much). We presented a total of ten images at random in this section to lessen information bias. Eight images were related to alcohol companies and the rest were unrelated brands. Similarly, the participants were given a maximum of ten seconds to react to each image.

### 2.5. Primary Independent Variable—Types of Alcohol Logo

We categorized logos into five types to study the effect of each type of logo: (1) original logo; (2) black logo; (3) partial logo; (4) logo on non-alcoholic beverage with the package; and (5) logo on other merchandise. The logos were modified to meet our requirements for each type. Alcoholic logos used in this study were mostly from beer lineups. Our pool of logos included logos from products that were related and unrelated (i.e., soft drink brands, banks) to alcoholic companies.

-Original logo was the original brand mark used in real-life settings (Figure 1a).-Black logo was the original logo in which the original color was changed to black (Figure 1b).-Partial logo was the original logo with some parts removed (Figure 1c).-Non-alcoholic beverage with the package (e.g., drinking water, soda) of the alcoholic brand that used a similar logo as the alcoholic products (Figure 1d).-Logo on other merchandise was the original logo that appears on merchandise other than the beverages (Figure 1e).

### 2.6. Other Variables

Our data collection included gender (female and male), age, area of residence (southern, central, northern, and northeastern regions), education level (uneducated, primary school, secondary to high school, diploma, bachelor’s degree, and master’s degree or above), monthly household income in THB (USD 1 ~ THB 33), and desire to drink alcohol at the time of survey. We categorized the age of our participants into two categories (i.e., younger, aged 19–24 years, and older, aged >25 years. We also categorized the household income into four categories (i.e., THB <5000, 5000–10,000, 10,001–20,000, and >20,000).

### 2.7. Statistical Analysis

Descriptive data were presented as frequencies with percentage and as median with standard deviation (SD). We used the linear mixed-effects models to compare the differences in their reaction times and the desire to drink alcohol of each type of logo of each participant. These models were chosen given the hierarchical nature of the data with clustering of reaction times and desire to drink of logos (lower or first level) within participants (higher or second level). We conducted a multiple linear mixed-effects regression analysis of the reaction time and scores of the desire to drink alcohol to adjust for possible confounders (i.e., gender, age group, area of residence, education level, ASSIST score, and the desire to drink at the time of the survey) that could affect the reaction time and desire to drink [32]. The final models were fitted by maximum likelihood and met the assumptions. We also applied the subgroup analysis by the various logos. There were six models of brand recognition time and six models of desire to drink. R-software version 4.2.1 [33] and MASS packages [34] were used for data analysis.

### 2.8. Ethical Considerations

Online consent was acquired at the beginning of the questionnaire. The survey was anonymized; however, participants who participated in the prize draw gave their phone number. The prize was THB 200 worth of Starbucks gift cards for 60 participants. This study was approved by the Faculty of Medicine Human Research Ethic Committee (HREC), Prince of Songkla University (REC. 63-367-9-6).

## 3. Results

### 3.1. Demographic Characteristics of the Participants

The study included 1185 participants with an average age of 36.4 years and a standard deviation of 13.4. The majority (65.8%) of our participants were female, 43.9% lived in southern Thailand, and 48.6% held a bachelor’s degree. Participants who worked in the government sector were represented by 23.8%, while 47.1% had a household income more than THB 20,000 ~ USD 606 (USD 1 ~ THB 33) per month, 76.0% had a low risk of alcohol addiction, and 88.3% did not have a desire to drink alcohol at the time of the survey (Table 1).

### 3.2. Median and Multivariate Regression Analyses of the Alcohol Recognition Using Reaction Time and Score on the Desire to Drink Alcohol

From the multivariate regression analysis (Table 2), the participants took a longer time to recognize the modified logos than the original logos. The estimated coefficients for the black logo, partial logo, other merchandise, and non-alcoholic beverage were 0.105 s (95% confidence interval (CI) 0.064, 0.147), 0.126 s (95% CI 0.084, 0.167), 0.479 s (95% CI 0.432, 0.527), and 0.405 s (95% CI 0.362, 0.448), respectively.

Compared to the original logo, the participants reported less desire to drink alcohol after seeing the black logo, non-alcoholic beverage, and logo on other merchandise, and the estimated coefficients were −0.149 (95% CI −0.232, −0.066), −0.415 (95% CI −0.498, −0.332), and −0.427 (95% CI −0.510, −0.343), respectively. We found no significant difference in the median scores in the desire to drink between each logo type. Moreover, no difference was observed in the estimated coefficients in the desire to drink between the original logo and the partial logo.

### 3.3. Subgroup Analysis of Alcohol Recognition Using Reaction Time to Each Type of Logo

Based on the ASSIST score, participants in the high-, intermediate-, and low-risk groups had estimated coefficients that were −0.38 (95% CI −0.67, −0.08), −0.42 (95% CI −0.6, −0.24), and −0.41 (95% CI −0.56, −0.26), respectively, who reacted significantly faster to the original logos compared to the non-drinker group (Table 3). Interestingly, the participants in these three groups were reminded of alcoholic products after seeing non-alcoholic beverages at speeds that were not significantly different. Moreover, participants in the young age group significantly recognized the alcohol at a faster rate than participants aged 25 years old or older.

### 3.4. Subgroup Analysis of the Score of the Desire to Drink after Seeing Each Type of Logo

In all logo categories, the high-, intermediate-, and low-risk groups obtained significantly higher scores in the desire to drink than the non-drinker group (Table 4). Similarly, participants who did not desire to drink alcohol at the time of the survey had significantly less desire to drink alcohol after being shown all types of logos.

## 4. Discussion

### 4.1. Alcohol Recognition Using Reaction Time

We aimed to assess the difference in alcohol recognition using reaction time and desire to drink among consumers exposed to different brand images. We found that the participants reacted differently to each type of logo. They were more sensitive to the original logos, but logos that were modified into different forms could trigger their recognition of alcoholic beverages. According to the results, we could observe statistical significance in the increase of reaction time when changing the color of a logo to black. To our knowledge, this is the first study to assess the alcohol recognition of surrogate logos using reaction time. However, the estimated coefficient was less than 0.5 s. Thus, this did not show practical significance. Similarly, when participants saw the whole package of non-alcoholic products that could be seen in various situations in daily life (e.g., supermarkets, restaurants), they were reminded of the alcoholic products with statistical significance. However, this was not practically significant as the estimated coefficient of reaction was less than 0.5 s. Thus, it could be inferred that those non-alcoholic products were able to represent their alcoholic counterparts. Yang et al. reported that the evaluation of brand extension was related to the emotional process that involved insular activity, and the people made evaluations more quickly when they were opposed to the extension [35]. Our results, showing that participants could recognize modified logos as alcohol, were in line with the previous findings of the literature [19,27,28].

### 4.2. Desire to Drink Alcohol

The participants had a desire to drink alcohol even when presented with black logos, the non-alcoholic beverages, and other merchandise with the original logos. However, it is necessary to point out that the estimated coefficient in the desire to drink compared to the original logo was less than the original logo, by less than 0.5 points in a scoring range from 0 to 4. Although this was statistically significant, it was almost the same reaction time between the original and modified logos in terms of practical significance (less than 0.5 points). Our results support a previous study in India that found that a surrogate product could influence consumers to purchase a particular product [15]. Similarly, considering our results demonstrated that the original logos and other modified logos gave comparable scores in the desire to drink, it can be concluded that the modified logos could contribute to the purchase of alcohol in the same way as the original images of the products. Therefore, the handling of modified logos should be no less different from the original ones.

### 4.3. Implications for the Youth and Females

This study found that younger participants aged 19–24 years recognized the logos significantly faster than the older participants (>25 years) and had almost the same score of desire to drink alcohol between the original and modified logos. This finding highlights the concern of the impact of surrogate marketing on the youth in Thailand. Although there is no proof that alcohol marketing can initiate alcohol consumption in the youth [36], our results could provide additional evidence of the impact of marketing on the youth. Although our data indicated that female participants reacted more slowly and had less desire to drink than men, it is important to keep an eye on the alcohol industry as more marketing tactics focus on females [37].

### 4.4. Theoretical Contribution from Our Study

There was no direct question of how our participants evaluated the extensions in our study; those who had no desire to drink alcohol during the survey had longer reaction times than those who did. Nonetheless, we could say that the differences in the reaction times toward each type of logo may have arisen from differences in the emotional process of each person. However, based on our results, we found that the two components of alcohol brand recognition are the color and shape of original logos. The locations of the logo on the other packaging or merchandise are not associated with alcohol brand recognition and desire to drink.

### 4.5. Policy Implication for a Policy against Surrogate Marketing in Thailand and Other Countries

This study found that the participants reacted to original logos and modified logos in almost the same reaction time, by less than 0.5 s. Regarding the desire to drink, the desire to drink scores between original and modified logos were almost the same. These results may infer that modified logos could function as substitutions for the original logos in the minds of consumers. For practical contribution based on our results, we suggest that the policy-makers (i.e., the Office of Alcohol Control Committee, Department of Disease Control, Ministry of Public Health, Thailand) in Thailand and other countries place a total ban on all surrogate logos. Logos that are designed to look similar to the alcoholic logos by either changing the color or using part of the original logos should be prohibited. The new trademark registration of such logos should not be allowed. As for the current modified logos, there should be restrictions on advertisements that may serve directly or indirectly as a reference for alcoholic products.

### 4.6. Strengths and Limitations

The strength of our study lies in the fact that little research has been published on brand extension and its effect on consumer alcohol brand recognition using CAPI to evaluate reaction times. This study opens a new door regarding how fast a person can recognize a logo as well as different reaction times for several types of logos. Moreover, this study moves beyond recognition to determine whether or not seeing an extended brand mark can trigger a desire in a person to drink alcohol.

It is also important to note the limitations of our study. We chose an online platform as it was able to reach a large number of people in a wide geographic area in a short period of time, which was especially crucial in the COVID-19 era. However, selection bias should be considered since the participants were only those who could use the internet and have access to the social media that we used to distribute our survey. The participants with this background might have reacted faster to the alcohol recognition section because they were more familiar with the online platform than the general population. Nonetheless, the results of the desire to drink might not be different between our sample and the general population. Furthermore, most participants in our study population were in the age range of 20–39 with a high level of education, which does not represent the general population in Thailand.

Since we aimed to evaluate the effects of alcohol extended logos, we modified the logos into versions that might not be available in the real world. We acknowledge that advertisements in the real world use extended brand marks that are more complex with accompanying context that might easily trigger brand recognition and the desire to drink (e.g., a picture showing a party setting). Future studies may look into these real-world advertisements to explore the extent to which surrogate marketing affects consumers. Moreover, as the alcoholic products included in our studies were mainly central products that brands often used in surrogate marketing, the alcoholic products were mostly beer. We suggest that future studies should include a larger variety of products to give a deeper idea of how alcoholic logos could affect brand recognition.

## 5. Conclusions

Consumers in Thailand constantly face surrogate marketing of alcohol companies in various forms. It was clear that the modified logos (i.e., black logos, partial logos, logos on non-alcoholic beverages and other merchandise) used in this study could remind consumers of the alcohol products and even bring about the desire to drink the beverages. Moreover, it is recognized that the youth are affected by schemes as much as older persons in terms of the desire to drink alcohol, and also, since they react faster to all types of logos, we need to protect our youth from surrogate marketing of alcohol products. The public and policy-makers must be made aware of this marketing scheme and the potential harm to the general population. Furthermore, we recommend concrete policy changes to control surrogate marketing of alcohol products.

## Figures and Tables

**Figure 1 ijerph-19-11756-f001:**
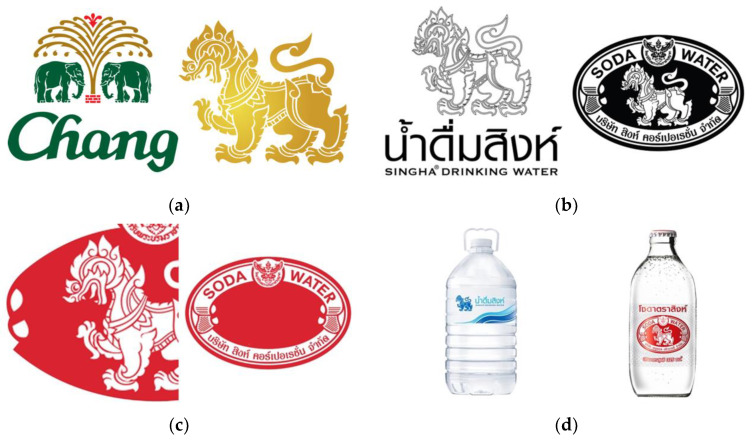
Examples of alcohol-related logos used in the survey: (**a**) original logo of beer; (**b**) black logo of drinking and soda water; (**c**) partial logo of soda water; (**d**) non-alcoholic beverage with package of drinking and soda water; and (**e**) logo on other merchandise.

**Table 1 ijerph-19-11756-t001:** Sociodemographic characteristics of the participants (*n* = 1185).

Variable	*n* (%)
Gender	
Female	779 (65.8)
Male	405 (34.2)
Age (years), mean (SD)	36.4 (13.4)
Age groups	
Younger (19–24 years)	295 (25.0)
Older (>25 years)	887 (75.0)
Area of residence in Thailand	
Southern	520 (43.9)
Central	473 (39.9)
Northern	153 (12.9)
Northeastern	39 (3.3)
Education level	
Uneducated	1 (0.1)
Primary school	25 (2.1)
Secondary to high school	162 (13.7)
Diploma	60 (5.1)
Bachelor’s degree	574 (48.6)
Master’s degree or above	360 (30.5)
Occupation	
Student	292 (24.6)
Government sector	282 (23.8)
Private sector employee	206 (17.4)
Business	114 (9.6)
Retired	47 (4.0)
Agriculture	41 (3.5)
Job hunting and others	107 (9.0)
Monthly household income (THB)	
<5000	150 (12.7)
5000–10,000	260 (22.0)
10,001–20,000	215 (18.2)
>20,000	557 (47.1)
ASSIST score	
High risk	41 (3.5)
Intermediate risk	241 (20.5)
Low risk	724 (61.7)
Non-drinkers	168 (14.3)
Desire to drink alcohol at the time of survey	
No	1046 (88.3)
Yes	139 (11.7)

SD, standard deviation; THB, Thai baht (USD 1 ~ THB 33); ASSIST, alcohol, smoking, and substance involvement screening test.

**Table 2 ijerph-19-11756-t002:** Median and multivariate regression analyses on alcohol recognition using reaction times and scores on the desire to drink.

Variables	Reaction Time, Seconds			Desire to Drink, Points		
	Median (IQR)	Estimated Coefficient (95% CI)		Median (IQR)	Estimated Coefficient (95% CI)	
		Unadjusted ^1^	Adjusted ^1^		Unadjusted ^1^	Adjusted ^1^
Original logo	1.4 (1.1, 1.8)	Ref.	Ref.	1 (0, 3)	Ref.	Ref.
Modified logo						
Black logo	1.4 (1.2, 1.9)	0.094 * (0.046, 0.142)	0.105 * (0.064, 0.147)	0 (0, 2)	−0.133 (−0.207, −0.0588)	−0.149 * (−0.232, −0.066)
Partial logo	1.4 (1.2, 1.9)	0.114 * (0.066, 0.162)	0.126 * (0.084, 0.167)	1 (0, 2.2)	0.0005 (−0.0738, 0.0749)	−0.005 (−0.088, 0.078)
Non-alcoholic beverage	1.6 (1.3, 2.3)	0.438 * (0.390, 0.486)	0.479 * (0.432, 0.527)	0 (0, 2)	−0.0329 * (−0.404, −0.255)	−0.415 * (−0.498, −0.332)
Other merchandise	1.6 (1.3, 2.3)	0.405 * (0.357, 0.453)	0.405 * (0.362, 0.448)	0 (0, 2)	−0.0362 * (−0.436, −0.287)	−0.427 * (−0.510, −0.343)

^1^ Linear mixed-effects model fit by maximum likelihood; model adjusted for gender, age group, area of residence, education level, the alcohol, smoking, and substance involvement screening test (ASSIST), and the desire to drink alcohol at the time of the survey. IQR, interquartile range; CI, confidence interval; * *p* value < 0.05.

**Table 3 ijerph-19-11756-t003:** Subgroup analysis of alcohol recognition using alcohol recognition using reaction time to each type of logo.

Variables	Estimated Coefficients of Reaction Times, Seconds (95% CI) ^1^					
	Original Logo	Modified Logo				
		Black Logo	Partial Logo	Non-Alcoholic Beverage	Other Merchandise	All Modified Logo
	Model 1	Model 2	Model 3	Model 4	Model 5	Model 6
Intercept	2.23 (1.99, 2.48) *	2.31 (2.06, 2.57) *	2.19 (1.94, 2.44) *	2.44 (2.16, 2.72) *	2.44 (2.17, 2.71) *	2.35 (2.14, 2.55) *
Gender (ref. = Male)						
Female	0.03 (−0.07, 0.13)	0.1 (0, 0.21)	0.02 (−0.09, 0.12)	0.01 (−0.11, 0.13)	−0.02 (−0.13, 0.09)	0.03 (−0.06, 0.11)
Age group (ref. =>25 years old)						
Young group (19–24 years)	−0.36 (−0.48, −0.23) *	−0.36 (−0.49, −0.24) *	−0.32 (−0.45, −0.19) *	−0.38 (−0.52, −0.24) *	−0.34 (−0.48, −0.21) *	−0.35 (−0.46, −0.25) *
Area of residence in Thailand (ref. = Central)						
Northeastern	0.13 (−0.14, 0.40)	0.23 (−0.05, 0.51)	0.15 (−0.12, 0.43)	0.23 (−0.08, 0.54)	0.23 (−0.07, 0.53)	0.21 (−0.01, 0.43)
Northern	0.12 (−0.04, 0.27)	0.14 (−0.02, 0.30)	0.11 (−0.05, 0.27)	0.02 (−0.16, 0.20)	0.05 (−0.12, 0.22)	0.08 (−0.05, 0.21)
Southern	0.18 (0.06, 0.29) *	0.19 (0.07, 0.31) *	0.16 (0.04, 0.27) *	0.07 (−0.06, 0.20)	0.11 (−0.02, 0.23)	0.13 (0.03, 0.23) *
Education level (ref. = Uneducated, primary school to high school)						
Diploma	−0.06 (−0.30, 0.19)	−0.33 (−0.58, −0.07) *	−0.2 (−0.45, 0.05)	−0.09 (−0.37, 0.20)	−0.07 (−0.34, 0.20)	−0.17 (−0.38, 0.03)
Bachelor’s degree	−0.36 (−0.50, −0.22) *	−0.36 (−0.51, −0.21) *	−0.31 (−0.46, −0.17) *	−0.35 (−0.52, −0.19) *	−0.28 (−0.44, −0.12) *	−0.33 (−0.45, −0.21) *
Master’s degree or above	−0.48 (−0.64, −0.32) *	−0.47 (−0.63, −0.3) *	−0.32 (−0.49, −0.16) *	−0.37 (−0.56, −0.18) *	−0.35 (−0.53, −0.18) *	−0.38 (−0.51, −0.24) *
ASSIST score (ref. = Non-drinkers)						
High risk	−0.38 (−0.67, −0.08) *	−0.38 (−0.69, −0.08) *	−0.21 (−0.51, 0.09)	−0.15 (−0.49, 0.19)	−0.16 (−0.49, 0.16)	−0.23 (−0.47, 0.02)
Intermediate risk	−0.42 (−0.60, −0.24) *	−0.34 (−0.52, −0.15) *	−0.24 (−0.43, −0.06) *	−0.04 (−0.25, 0.17)	−0.22 (−0.42, −0.02) *	−0.21 (−0.36, −0.06) *
Low risk	−0.41 (−0.56, −0.26) *	−0.29 (−0.45, −0.14) *	−0.22 (−0.38, −0.07) *	−0.11 (−0.28, 0.07)	−0.2 (−0.36, −0.03) *	−0.21 (−0.33, −0.08) *
Desire to drink alcohol at the time of survey (ref. = Yes)						
No	0.21 (0.06, 0.37)	0.07 (−0.09, 0.23)	0.15 (−0.01, 0.3)	0.21 (0.04, 0.39) *	0.23 (0.06, 0.4) *	0.16 (0.04, 0.29) *

^1^ Linear mixed-effects model fit by maximum likelihood. * *p* value < 0.05. CI, confidence interval; ASSIST, alcohol, smoking, and substance involvement screening test.

**Table 4 ijerph-19-11756-t004:** Subgroup analysis of score of desire to drink after seeing each type of logo.

Variables	Estimate Coefficient of Desire to Drink, Points (95% CI) ^1^					
	Original Logo	Modified Logo				
		Black Logo	Partial Logo	Non−Alcoholic Beverage	Other Merchandise	All Modified Logo
	Model 1	Model 2	Model 3	Model 4	Model 5	Model 6
Intercept	1.11 (0.73, 1.49) *	1.55 (1.15, 1.95) *	1.71 (1.29, 2.12) *	1.15 (0.74, 1.56) *	1.43 (1.05, 1.81) *	1.48 (1.17, 1.8) *
Gender (ref. = Male)						
Female	−0.21 (−0.38, −0.03) *	−0.29 (−0.46, −0.12) *	−0.3 (−0.47, −0.12) *	−0.04 (−0.21, 0.13)	−0.23 (−0.4, −0.07) *	−0.22 (−0.35, −0.08) *
Age group (ref. =>25 years)						
Young (19–24 years)	−0.03 (−0.23, 0.17)	−0.03 (−0.23, 0.17)	0.02 (−0.19, 0.23)	−0.06 (−0.27, 0.14)	−0.15 (−0.34, 0.04)	−0.06 (−0.22, 0.1)
Area of residence in Thailand (ref. = Central)						
Northeastern	−	0.14 (−0.3, 0.59)	0.26 (−0.2, 0.73)	−0.22 (−0.67, 0.23)	0.14 (−0.28, 0.56)	0.07 (−0.28, 0.42)
Northern	−	−0.01 (−0.26, 0.25)	0.08 (−0.18, 0.34)	−0.19 (−0.45, 0.06)	0.12 (−0.12, 0.36)	0 (−0.2, 0.2)
Southern	−	−0.3 (−0.49, −0.11) *	−0.28 (−0.48, −0.09) *	−0.1 (−0.29, 0.09)	−0.21 (−0.38, −0.03) *	−0.23 (−0.38, −0.08) *
Education level (ref. = Uneducated, primary school to high school)						
Diploma	0.27 (−0.14, 0.68)	0.05 (−0.36, 0.46)	0.1 (−0.32, 0.52)	0.23 (−0.17, 0.64)	0.01 (−0.38, 0.41)	0.1 (−0.22, 0.42)
Bachelor’s degree	0.39 (0.15, 0.62) *	0.27 (0.03, 0.5) *	0.13 (−0.12, 0.37)	0 (−0.24, 0.24)	0.04 (−0.18, 0.27)	0.11 (−0.08, 0.29)
Master’s degree or above	0.34 (0.08, 0.6) *	0.27 (0.01, 0.53) *	0.12 (−0.15, 0.39)	−0.03 (−0.3, 0.24)	0.01 (−0.24, 0.27)	0.09 (−0.12, 0.3)
ASSIST score (ref. = Non-drinkers)						
High risk	1.86 (1.39, 2.33) *	1.55 (1.05, 2.04) *	1.46 (0.96, 1.97) *	1.25 (0.75, 1.74) *	1.04 (0.57, 1.51) *	1.31 (0.92, 1.69*)
Intermediate risk	1.42 (1.13, 1.71) *	0.97 (0.67, 1.26) *	1.12 (0.81, 1.43) *	0.66 (0.36, 0.96) *	0.69 (0.41, 0.97) *	0.84 (0.61, 1.08) *
Low risk	0.67 (0.43, 0.91) *	0.45 (0.21, 0.7) *	0.53 (0.28, 0.79) *	0.27 (0.02, 0.52) *	0.31 (0.08, 0.54) *	0.39 (0.19, 0.58) *
Desire to drink alcohol at the time of survey (ref. = Yes)						
No	−0.80 (−1.06, −0.54) *	−0.87 (−1.12, −0.62) *	−0.90 (−1.16, −0.63) *	−0.43 (−0.68, −0.17) *	−0.68 (−0.92, −0.44) *	−0.73 (−0.93, −0.53) *

^1^ Linear mixed-effects model fit by maximum likelihood. * *p* value < 0.05. CI, confidence interval; ASSIST, alcohol, smoking, and substance involvement screening test.

## Data Availability

Not applicable.

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
