# Peer review of "Alcohol Recognition and Desire to Drink of Extended Alcohol Brand Logos"

_ijerph, 2022, doi:10.3390/ijerph191811756_

Round 1

Reviewer 1 Report

In the introductory part the CAPI is mentioned, however in the methodological part the descriptions mentioned on-line interview. The authors have to provide clear information about the way, the questionaires were collected. The authors also have to add some information about measurement the reaction time and the instructions they gave to the respondents as well as how the respondents get information abou the research. 

Reviewer 2 Report

The paper is interesting, however, there are some comments for further improvement. 

First, regarding the title, the "cross-sectional study" implies limitation, the authors are recommended to remove the "cross-section study" in the title. 

Second, regarding the abstract, an abstract using paragraph is good enough, please remove the heading "introduction", "methods", etc. 

Third, social-media marketing is important for alcohol, especially for the social-media influencer marketing, because social-media influencers exert their influence by transferring their meaning to the alcohol and thereby persuading consumers to adopt their recommendation. Thus, the authors are strongly recommended to add the following references in the introduction and literature review section to highlight the importance of social-media marketing: 

Westgate, E. C., & Holliday, J. (2016). Identity, influence, and intervention: the roles of social media in alcohol use. Current Opinion in Psychology9, 27-32.

Cheung, M. L., Leung, W. K., Aw, E. C. X., & Koay, K. Y. (2022). “I follow what you post!”: The role of social media influencers’ content characteristics in consumers' online brand-related activities (COBRAs). Journal of Retailing and Consumer Services66, 102940.

Roberson, A. A., McKinney, C., Walker, C., & Coleman, A. (2018). Peer, social media, and alcohol marketing influences on college student drinking. Journal of American college health66(5), 369-379.

Reviewer 3 Report

I cannot give any recommendations. Generally, the paper has everything in its place but is weak and as a whole is not ready for presentation. That is my subjective feeling. That paper could be accepted in a weaker Journal without IF and it will be OK, but now is too weak for IJERPH quality.

Reviewer 4 Report

The paper approaches an interesting subject that is relevant for the journal.

However a number of changes are needed for the improvement of the paper. Here are some general, as well as punctual suggestions:

- you need to include a section of literature review in which to explain the theories that support your research

- you need a section that discusses the alcohol consumption in Thailand (statistics, cultural influencing factors, consumption habits, policies?)

- p. 1  line 36: explain "Thailand's alcohol advertising in recent years has become more complex and problematic"

- p. 2 line 50: " Previous studies ...." which studies? You need to name them (with bibliographical source)

- you need to define surrogate marketing and other concepts used in the paper

- you need to better explain the method used to set the sample size (by describing it first and then applying it)

- what king of prize draw was offered to potential respondents?

- the entire questionnaire needs to be included in an Appendix

- the logos used in the research were associated with what type of alcoholic products?

- you need to include/list in an appendix all logos included in the study (or brands): to what type of products do they belong? what type of variable do they represents?

- the logo example in the paper refer to what type of product?

- p. 4  line 130: "... to respond within ten seconds per image" Not clear how answers were interpreted: length for the answer? or within or outside the 10 seconds time?

- in the text specifically name the town and the country of the university involved in the study

- for the average income also transform it and give the amount in USD (at least in the text - p. 4 line 165).

- p. 9 lines 238 and 249 : " ...had almost the same..." - this is a comparison: almost the same to whom? Please clarify and re-phrase.

- you need to compare the results of your study with more previous studies on similar topics.

- p. 10, lines 265-268: be specific: whose policies are you referring to? who should implement restrictions? For whom? How does this fit to the political and legal systems in Thailand?

- what are the theoretical contributions of the paper?

- what are the practical contribution of the paper?For  whom?

- p. 11, line 303: what are the concrete policy changes you recommend?

- bibliographical sources are too limited.

Round 2

Reviewer 2 Report

The authors have done a good job in revising the manuscript. 

Reviewer 3 Report

-

Reviewer 4 Report

Authors have incorporated most of the reviewer's suggestions.

Paper gain clarity in content.